# ReGen4AD: Retrieval Based Online Video Generation for Reactive Autonomous Driving Simulation

## Abstract

We introduce **ReGen4AD**, an interactive and controllable retrieval based online video generation pipeline for closed-loop reactive driving evaluation. Unlike existing video generative models for AD which generate multiple frames all at once, the proposed designs are tailored for interactive simulation, where sensor rendering and behavior rollout are decoupled by applying a separate behavioral controller to simulate the reactions of surrounding agents. As a result, the generative model could focus on image fidelity, control adherence, and spatial-temporal coherence. For temporal consistency, due to the stepwise interaction nature of simulation, we design a noise modulating temporal encoder with Gaussian blurring to encourage long-horizon autoregressive rollout of image sequences without deteriorating distribution shifts. For spatial consistency, a retrieval mechanism, which takes the spatially nearest images as references, is introduced to ensure scene-level rendering fidelity. The spatial relations between target and reference are explicitly modeled with 3D relative position encodings. The potential over-reliance of reference images is mitigated with hierarchical sampling and classifier-free guidance. We compare the generation quality with existing AD generative models and show its superiority in the online driving setting. We further integrate it into nuPlan and evaluate the generative qualities with closed-loop simulation results.

## 1 Introduction

Deep generative models have witnessed fast development in vision (Song et al., 2022; Rombach et al., 2022b), and work has explored the application to autonomous driving (AD) e.g. for generating sensor data as augmentation for perception. Such efforts include generating static BEV-conditioned driving scenarios for detection and mapping (Gao et al., 2024a; Yang et al., 2023a; Swerdlow et al., 2024), producing video clips (Wen et al., 2023; 2024; Ma et al., 2024) for tracking and trajectory prediction. Another line of work utilizes video diffusion models as world models to implicitly simulate driving scenarios (Gao et al., 2024b; Wang et al., 2023b; Zhao et al., 2024).

For the evaluation of AD models, the aimed task for generative models in this paper, there are two types: open-loop (Caesar et al., 2020) and closed-loop (Dosovitskiy et al., 2017). The former typically measures displacement errors between predicted trajectories and logged expert trajectories. Recent studies (Zhai et al., 2023; Li et al., 2024; Dauner et al., 2023) reveal that it suffers from imbalanced datasets, heavy reliance on expert ego states, and distribution shift, etc. While the closed-loop ones (Prakash et al., 2021; Chitta et al., 2023; Jia et al., 2024b) rely on computer graphics engines, e.g. the popular CARLA (Dosovitskiy et al., 2017) platform. The engine evaluates the planning performance in a reactive way, but there still remain notable gaps between simulator and real world from both rendering and behavioral perspectives.

A natural idea is to provide real-world sensor data to the simulation environment with generative models. Fig. 1 gives a comparison of existing generative paradigms for AD. Frame-wise image generation (Swerdlow et al., 2024; Yang et al., 2023a; Gao et al., 2024a) lacks the constraint of temporal consistency. Video-level generation (Wen et al., 2024; Ma et al., 2024; Wang et al., 2023b) relies on layout control sequences at future timesteps, which is unable to perform step-by-step interactions with E2E-AD models. Predictive video generation (Gao et al., 2024b; Wang et al., 2023c; Hu et al., 2023a) directly outputs images in pixel space, making it hard to collect information for calculating closed-loop driving metrics.

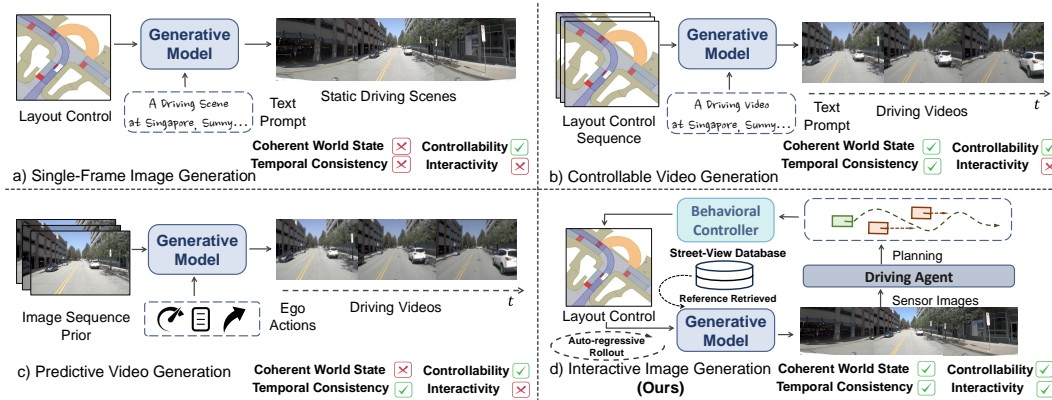

Figure 1: **Different Paradigms of Generative Models for Autonomous Driving**: (a) **Single-Frame Image Generation** (Swerdlow et al., 2024; Yang et al., 2023a; Gao et al., 2024a), as relatively early works, do not account for temporal generation. (b) **Controllable Video Generation** (Wen et al., 2024; Ma et al., 2024; Wang et al., 2023b) focuses on generating videos with controls for each frame, which is not suitable for interactive simulation. (c) **Predictive Video Generation** (Gao et al., 2024b; Wang et al., 2023c; Hu et al., 2023a) emphasizes the annotation-free training ability, which lacks the ability to adhere to control. (d) **Interactive Image Generation**: The proposed framework leverages the power of generative models in an autoregressive manner, enabling high-frequency interactions with end-to-end driving models and generating temporally consistent images. The integration of a rule-based behavioral controller simulates the behavior of other driving agents to ensure a coherent world state and provides layout controls for the generative part of the framework.

To fill the gap above, we propose a novel generative model for interactive video generation to support closed-loop E2E-AD evaluation. Specifically, a simulation-oriented generative renderer capable of auto-regressively generating image sequences with controllability and spatial-temporal consistency is proposed. For **controllability**, we incorporate projected layout controls into the generation process. For **temporal consistency**, the previous frame could always be used as conditions to provide priors. However, it also introduces significant train-val gaps, which easily collapse generation due to the cumulative errors of autoregressive generation. We propose a noise modulation module with Gaussian blurring during the training process to adapt the model to the defective conditions. For **spatial consistency**, our observation is that **the static background could be retrieved from the database and thus the uncertainty is eliminated.** This point is quite different from video generation models since their purpose is to provide diverse samples while our goal is for high-fidelity simulation. Thus, we retrieve the two frames with lowest distance in the forward and backward directions respectively and use them as conditions to guide the generation of static background.

We compare ReGen4AD with SOTA generative models for AD and show its superior fidelity. In addition, we integrate ReGen4AD into a widely used planning benchmark -nuPlan (Karnchanachari et al., 2024) and develop a closed-loop interactive end-to-end simulation platform. We compare ReGen4AD with baseline generative methods by comparing the performance of the same E2E-AD models under different renderers, and ReGen4AD demonstrates explicit advantages. We conduct ablation studies and case analysis to evaluate the effectiveness of proposed modules. **Our contributions are:**

- We propose **ReGen4AD**, a simulation-oriented generative framework providing controllable and spatial-temporal consistent sensor image sequences. To our best knowledge, this is the first closed-loop video generator for reactive AD simulation.
- Different from the recent efforts (Gao et al., 2024b; Kim et al., 2021; Zhao et al., 2024; Hu et al., 2023a; Jia et al., 2023a; Wang et al., 2023c;b; Li et al., 2023b; Bandarupalli et al., 2024; Wen et al., 2024; 2023; Huang et al., 2024; Wu et al., 2024a) of video generation, we propose several simulation-oriented designs to enhance the fidelity, including techniques to avoid collapse of autoregressive process and a retrieval augmented generation paradigm able to utilize the offline database.
- We integrate ReGen4AD with an open-sourced planning simulator nuPlan and develop a closed-loop reactive simulation framework to empower closed-loop evaluation.

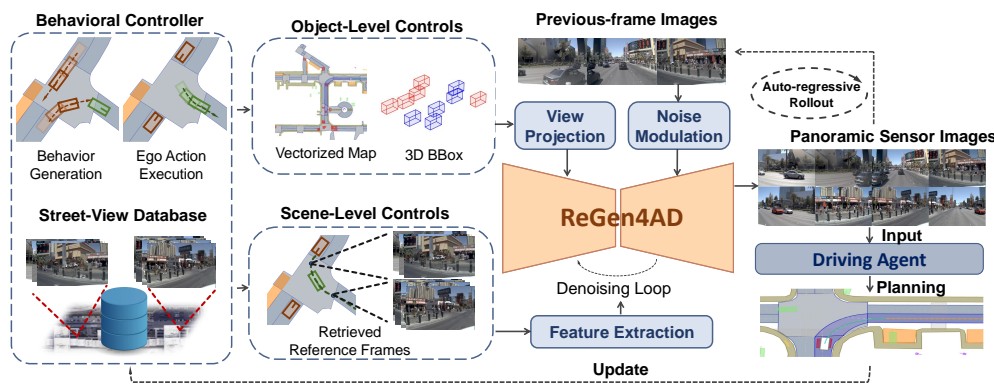

Figure 2: **Overall Framework**: The proposed ReGen4AD empowers closed-loop end-to-end evaluation by incorporating a behavioral controller to execute ego actions and generate behaviors of other driving agents. To improve fidelity of multi-view sensor images in an autoregressive manner, ReGen4AD (1) utilizes previous-frame image for temporal consistency; (2) retrieves spatially nearest reference image pair for background prior; (3) adheres to projected layout element controls for object-level consistency.

- Extensive experiments including ablation studies on the nuScenes and nuPlan datasets show the effectiveness of the whole proposed framework as well as each module.

## 2 RELATED WORKS

**Generative Models for AD and Robotics.** Diffusion models (Song et al., 2022; Ho et al., 2020) generate images by progressively denoising a randomly sampled Gaussian noise. Recent advances in the fields allow diffusion models to generate photorealistic synthesis of images conditioned on various inputs including text prompts, images, etc. (Zhang et al., 2023; Mou et al., 2023)

In AD, many works try to synthesize novel street-views with generative models. A line of works focuses on using generated images as data augmentation for downstream perception tasks. Some works (Xie et al., 2023; Zhou et al., 2024; Yang et al., 2023a; Swerdlow et al., 2024; Gao et al., 2024a; Zhang et al., 2024) use bounding boxes and map polylines as controls to generate single-frame driving scenes. Other works probe into layout controlled video generation with high temporal consistency (Li et al., 2023b; Bandarupalli et al., 2024; Wen et al., 2024; 2023; Huang et al., 2024; Wu et al., 2024a; Guo et al., 2024;?; Liang et al., 2023; Wu et al., 2024b; Deng et al., 2024) and explore the impact of synthesized data on detection (Li et al., 2023a; Zhu et al., 2024) and prediction (Jia et al., 2022; 2023b; 2024a) tasks. However, image generation model cannot guarantee frame-wise consistency while video generation models can't perform frame-by-frame interaction with the simulation environment.

Another line of work (Gao et al., 2024b; Zhao et al., 2024; Hu et al., 2023a; Jia et al., 2023a; Wang et al., 2023c;b; Kim et al., 2021; Wang et al., 2024a) focuses on the predictive capacity of generative models. They explore the possibility of turning video diffusion model into generalizable driving world model of a fully differentiable driving simulator, which rollouts world states in pixel space based on input actions and behaviors. However, such world models cannot guarantee a coherent world state and fail to provide interactive interfaces for simulations.

Some works in robotics (Wang et al., 2024b; Hua et al., 2024) also use generative models to collect offline manipulation data for imitation learning, which is distinct from our efforts to establish an online simulation platform.

Due to the above reasons, existing generation paradigms for AD and robotics fail to guarantee interactability and temporal-spatial coherence, which are two salient characteristics of simulation-oriented driving-scene generation.

**Closed-loop Simulation and Benchmarks.** E2E-AD methods transform the entire AD system into a learnable network to directly optimize the planning performance. Many existing E2E models are evaluated on the open-loop nuScenes protocol (Caesar et al., 2020), where the displacement errors between expert and predicted trajectories are used as metrics. However, open-loop evaluation has significant limitations, as pointed out in (Zhai et al., 2023; Li et al., 2024; Dauner et al.,

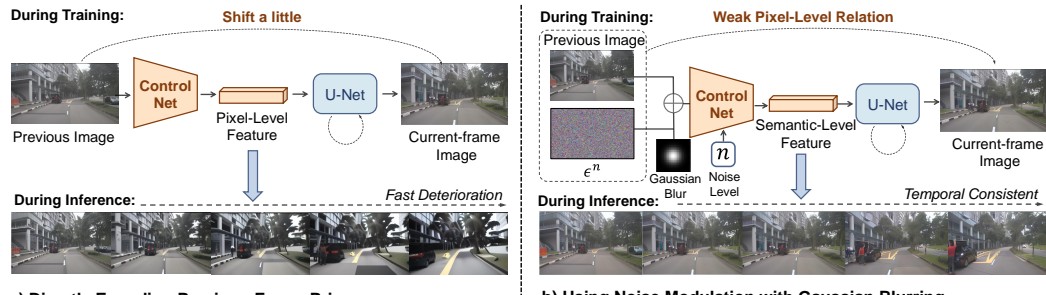

Figure 3: **Deal with Autoregressive Distribution Shift. a)** During training, due to the resemblance between previous and current frame prior, the model would overly rely on previous frame. During inference, the generation errors (artifacts) will cumulate and finally collapse. **b)** Adding Gaussian blurring and a random level of noise to previous images can destroy obvious pixel-level relations between the two frames. Noise level $n$ is fed as inputs to give model hints on the corrupted extent. As a result, the model can adapt to degenerated previous images and learn to extract high-level prior information instead of pixel-level copy.

2023). As for the closed-loop benchmarks, CARLA (Dosovitskiy et al., 2017)'s simulation has large gaps compared to real world at both rendering and behavior level while Waymax (Gulino et al., 2023) and nuPlan (Karnchanachari et al., 2024) are limited to bounding box level assessments. Recently, NAVSIM (Dauner et al., 2024) workarounds the problem by introducing an open-loop metric named PDMS that shows better correlation to close-loop metrics than displacement errors. However, NAVSIM assumes other vehicles are not reactive and fixes the ego vehicle's behavior in the simulation period, which cannot reflect the driving performance under highly interactive scenarios. Efforts are also spent on building closed-loop simulation with 3D reconstruction techniques (Yang et al., 2023b; Tian et al., 2024; Yang et al., 2024; Wu et al., 2024c). However, reconstruction methods are often spatially confined due to their heavy optimization costs, and suffer from weak generalizability for log-deviating trajectory.

In this work, we aim to build a closed-loop interactive end-to-end driving simulation, leveraging the recent huge advance in generative models.

## 3 METHODS

### 3.1 OVERALL FRAMEWORK

As shown in Fig. 2, we decouple the whole simulation framework into two parts: a behavioral controller for executing ego actions and propagating world states, and ReGen4AD, which unifies multiple simulation-oriented designs to achieve controllable and interactive surrounding image generation in AD scenarios based on structural information from behavioral controller and environment references retrieved from street-view database. At a given time $t$, ReGen4AD receives the following information:

1. **3D bounding boxes and semantic labels**: $\mathbf{B_t} = \{(b_i, c_i)\}_{i=1}^{N_b}$, where $b_i = (x_j, y_j, z_j)_{j=1}^8 \in \mathbb{R}^{8 \times 3}$ is the bounding boxes for both dynamic and static objects (cars, pedestrians, obstacles, etc.) within a specific range; $c_i \in \mathcal{C}_{box}$ is the semantic label.

2. **Vectorized map elements**: $\mathbf{M_t} = \{(v_i, c_i)\}_{i=1}^{N_m}$, where $v_i = (x_j, y_j)_{j=1}^{N_v}$ represents vertices for polygon map elements (roadblocks, cross-walk regions, etc.) and interior points for linestring map elements (lane dividers, etc.); $c_i \in \mathcal{C}_{map}$ represents the map class.

3. **Ego states**: $\mathbf{E_t} \in \mathbb{R}^{N_e}$, including ego velocity, steering angle, ego-to-global matrix etc.

4. **Camera parameters**: $\mathbf{K} = \{\mathbf{K}_i \in \mathbb{R}^{4 \times 4}\}_{i=1}^{N_{cam}}$, where $\mathbf{K}_i$ is the camera transformation matrix composed of intrinsic and extrinsic matrices that transforms points from LIDAR coordinate system to image coordinate system.

5. **Original recorded sensor images with global ego coordinates**: $\{(\text{coord}_i, \mathbf{I}_i)\}_{i=1}^{N_f}$, where $N_f$ is the number of total frames in current scenario; $\text{coord}_i$ is the position of ego vehicle under global coordinate system; $I_i^{ref} \in \mathbb{R}^{N_{cam} \times C \times H \times W}$ represents sensor images collected by $N_{cam}$ cameras. Note that they are recordings of human driving and could

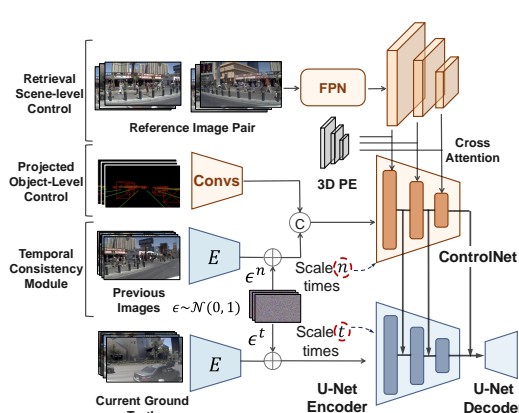

Figure 4: **Structural Design of ReGen4AD.** We design three additive modules to ensure controllable, consistent, and interactive image generation. **a) Temporal consistency module** incorporates previous frame images. Noise modulation module helps prevent distribution drift during autoregressive rollout. **b) Projected Object-Level Control** allows fine-grind controls over the location and orientation of driving vehicles in the scenario. **c) Retrieval Scene-Level Control** ensures spatial consistency by extracting multi-level features from nearest reference image pairs and injecting them into the ControlNet with attention mechanism.

not be directly used during simulation since the evaluated E2E-AD methods could behave differently from experts.

The detailed structure of ReGen4AD is illustrated in Fig 2. We will elucidate our simulation-oriented designs in the following sections. We also provide detailed introduction to LDM and ControlNet in the supplementary material.

## 3.2 TEMPORAL CONSISTENCY MODULE & MITIGATION OF AUTOREGRESSIVE DISTRIBUTION SHIFT

Unlike video diffusion models (Blattmann et al., 2023a;b; Ho et al., 2022), which improve temporal consistency by introducing attention along temporal axis of latent noise, ReGen4AD, as an autoregressive interactive generation method, improves temporal consistency by encoding previously generated images $\mathbf{I}_{t-1}$ with ControlNet (Zhang et al., 2023). Specifically, we encode $\mathbf{I}_{t-1}$ into latent space with the same VAE encoder as Stable Diffusion's and send it into the ControlNet encoder. The output hidden features from each layer of ControlNet are directly added to the corresponding layers of the U-Net encoder, as in Fig. 4.

However, utilizing the previous image introduces a train-val gap issue. During training, the previous images are always ground-truth while during inference, previous images are generated, which have gaps with real-world ones, though slightly. As a result, due to the recurrent nature of autoregressive generation, the error accumulates and could finally collapse the generation, as shown in Fig.3 (Left). It is called teacher-forcing (Lamb et al., 2016) or distribution shift (Ross & Bagnell, 2010) issue.

Since the deterioration stems from over-reliance on previous images (Valevski et al., 2024), we propose **noise modulation with Gaussian blur** to address the issue. Specifically, in training, previous-frame images are first encoded into latent space to get conditional previous latent $z_{\text{prev}}$. Then a random level of Gaussian noise is added to $z_{\text{prev}}$. The noise level is also input into the ControlNet encoder:

$$c_{\text{prev}} = \mathcal{E}(\sqrt{\bar{\alpha}_n}z_{\text{prev}} + \sqrt{1 - \bar{\alpha}_n}\epsilon \; ; \; n) \tag{1}$$

where $\mathcal{E}$ represents the ControlNet encoder; $\epsilon \in \mathcal{N}(0, 1)$ is the randomly sampled Gaussian noise and $n \in \mathcal{U}[0, N]$. The noise-adding policy is similar to the training strategy of diffusion models (Ho et al., 2020; Song et al., 2022), where the noise level $n$ here is analogous to the timestep $t$ in diffusion models. To further avoid accumulation of high-frequency artifacts, we apply Gaussian blurring to previous-frame images. As shown in Fig. 3 and later experiments, the proposed techniques could effectively alleviate the deterioration issues.

## 3.3 PROJECTED OBJECT-LEVEL CONTROL

This module follows existing generative models for AD (Wang et al., 2023b; Wen et al., 2023) and *we do not claim it as our contributions*. During simulation, the object-level information $\mathbf{B}_t$ and $\mathbf{M}_t$ is available from behavioral controller. Thus, to adopt them as control information, we project 3D bounding boxes $\mathbf{B}_t$ and vectorized map elements $\mathbf{M}_t$ from LIDAR coordinate system to 2D perspective view using the provided camera parameters $\mathbf{P}$ (Gao et al., 2024a; Wang et al., 2023b; Wen et al., 2023). Then we plot the projected discrete coordinates to form a set of binary masks of the same size with input images $\mathbf{B}_t^{\text{mask}} \in \mathbb{R}^{|\mathcal{C}_{\text{box}}| \times H \times W}$ and $\mathbf{M}_t^{\text{mask}} \in \mathbb{R}^{|\mathcal{C}_{\text{map}}| \times H \times W}$. We incorporate object semantic information by assigning each class its own dedicated channel. The two kinds of object level controls are concatenated and encoded into the latent space with a convolutional

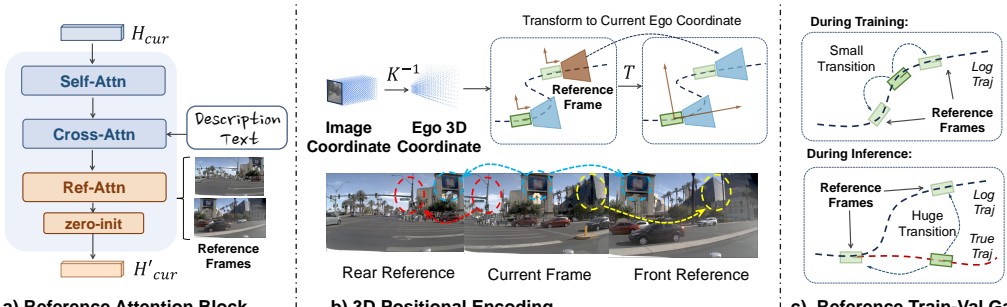

**Figure 5: Designs for Retrieval based Scene-Level Control. a)** Reference images are utilized in ControlNet with an additional cross attention module (Ref-Attn). **b)** Pixel-level 3D position encodings are calculated and fed into cross attention to provide spatial relations between reference and current images. **c)** E2E-AD agents might not follow logged trajectory during inference, leading to train-val gap.

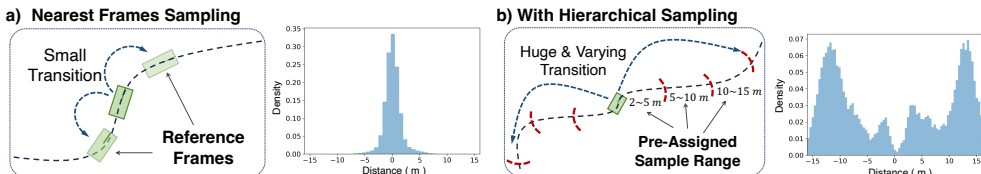

**Figure 6: Different Sampling Strategies for Reference Frames during Training. a)** Simply retrieving the closest one would make the model only able to deal with very similar reference images. **b)** The proposed hierarchical sampling strategy forces the model to adapt to reference images from a large scale of distance range.

network. Then, these encoded control signals are injected into the denoising process using the same ControlNet encoder described in Section 3.2. Our projected object-level control is:

$$c_{\text{proj}} = \mathcal{E}(\text{Conv}(\text{Cat}(\mathbf{B}_t^{\text{mask}}, \mathbf{M}_t^{\text{mask}}))) \tag{2}$$

### 3.4 REFERENCE-GUIDED SCENE-LEVEL CONTROL VIA IMAGE RETRIEVAL FROM OFFLINE DATABASE

Generative models are susceptible to creating fictitious artifacts (Decart et al., 2024). Previous studies on generative models for autonomous driving primarily focus on generating a diverse range of driving scenes (Gao et al., 2024a; Wen et al., 2023) to enhance the ground-truth dataset, effectively serving as a form of data augmentation. By contrast, **for the proposed simulation framework, fidelity is the most important factor for generative renderer since the tasks of enhancing diversity are assigned to the initial scenario selection and the behavioral controller**. In other words, we expect the diffusion models to faithfully follow controls and conditions, akin to a meticulous oil painter, rather than behaving like an artist who seeks to create diverse interpretations.

To achieve this, one key observation is that the background of scenes could be deterministically decided by referring recordings $\{(\text{coord}_i, \mathbf{I}_i)\}_{i=1}^{N_f}$ since the background is static. Thus, for high fidelity, extra control conditions could be applied by retrieval to eliminate uncertainty. Specifically, the two frames within recordings with closest distance (one from ahead of the ego vehicle and one from behind) to current location of ego agent are retrieved:

$$\mathcal{P} = \{(\mathbf{coord}_i - \mathbf{coord}_{\text{ego}}) \cdot \mathbf{v}_{\text{ego}}\}_{i=1}^{N_f}; \quad \mathbf{I}_{\text{ref}}^{\text{front}} = \mathbf{I}\left[\underset{i;\ \mathcal{P}_i > 0}{\arg\min}\ \mathcal{P}_i\right]; \quad \mathbf{I}_{\text{ref}}^{\text{rear}} = \mathbf{I}\left[\underset{i;\ \mathcal{P}_i < 0}{\arg\max}\ \mathcal{P}_i\right] \tag{3}$$

The two retrieved images are encoded by an image encoder (e.g., ResNet) and put into ControlNet as key and value in an additional cross-attention module so that the generation of current frames could find correspondence of background.

Further, since the spatial relations between current frame and the two reference frames could be explicitly calculated based on coordinate transformation, **we consider injecting pixel-wise spatial relations information into cross attention**. Specifically, pixel-wise 3D position encodings are adopted similarly to (Liu et al., 2022a;b). A discrete meshgrid $\mathbf{P}$ of size $(H, W, D, 4)$ in camera

Figure 7: **Case Study on Influence of Different Training Sampling Strategies.** Hierarchical sampling preserves generation quality under large deviation during inference.

Figure 8: **Effect for 3D Positional Encoding.**

Table 1: Comparisons of FID, 3D object detection and BEV segmentation on nuScenes validation set. The categorization of paradigms is based on Fig. 1.

| Paradigm | Method | FID↓ | FVD↓ | BEVFormer | | | BEVFusion (Camera Branch) | | | | StreamPETR | |
|---|---|---|---|---|---|---|---|---|---|---|---|---|
| | | | | NDS↑ | mAP↑ | mAOE↓ | NDS↑ | mAP↑ | mAOE↓ | mIoU↑ | NDS↑ | mAP↑ |
| | Original-nuScenes | - | - | 53.50 | 45.61 | 0.35 | 41.20 | 35.53 | 0.56 | 57.09 | 57.10 | 4 8.20 |
| Single-Frame Image Generation | BEVControl | 24.85 | - | 28.68 | 19.64 | 0.78 | - | - | - | - | - | - |
| | MagicDrive | 16.20 | 218.1 | 25.76 | 14.07 | 0.79 | 23.35 | 12.54 | 0.77 | 28.94 | 35.51 | 21.41 |
| Controllable Video Generation | DrivingDiffusion | 15.8 | 332.0 | - | - | - | - | - | - | - | - | - |
| | Panacea | 16.69 | 131 | 28.70 | 16.56 | 0.75 | 22.34 | 12.67 | 0.76 | 30.25 | 32.10 | 20.65 |
| | Panacea+ | 15.50 | 103 | 30.25 | 18.24 | 0.54 | 23.67 | 13.50 | 0.76 | 32.35 | 34.60 | 23.36 |
| Predictive Video Generation | DriveDreamer | 26.8 | 353.2 | 28.54 | 17.54 | 0.68 | 22.23 | 12.34 | 0.78 | 31.46 | 31.25 | 21.26 |
| | Drive-WM | 15.8 | 122.7 | - | - | - | - | - | - | - | - | - |
| Interactive Online Video Generation | ReGen4AD | **10.95** | **101** | **34.70** | **20.11** | **0.48** | **25.75** | **13.53** | 0.73 | **42.75** | **40.23** | **24.04** |

Table 2: Performance of UniAD's Different Tasks in nuScenes.

| Method | Detection | | BEV Segmentation | | | | Planning | | Occupancy |
|---|---|---|---|---|---|---|---|---|---|
| | NDS↑ | mAP↑ | Lanes↑ | Drivable↑ | Divider↑ | Crossing↑ | avg.L2(m)↓ | avg.Col.↓ | mIoU↑ |
| Original-nuScenes | 49.85 | 37.98 | 31.31 | 69.14 | 25.93 | 14.36 | 1.05 | 0.29 | 63.7 |
| MagicDrive | 29.35 | 14.09 | 23.73 | 55.28 | 18.83 | 6.57 | 1.18 | 0.33 | 54.6 |
| Panacea+ | 30.28 | 14.15 | 22.60 | 54.32 | 20.27 | 6.87 | 1.23 | 0.33 | 53.2 |
| ReGen4AD | **33.04** | **15.16** | **25.5** | **56.53** | **21.27** | **8.67** | **1.15** | **0.31** | **55.5** |

frustum space is calculated for all images, where $D$ is the number of points sampled along the depth axis. Then, the meshgrid is transformed from camera frustum space to current ego coordinate system with camera transformation matrix $\mathbf{P}^{\text{lidar}} = \mathbf{K}^{-1}\mathbf{P}$ for both current frame $\mathbf{P}_{\text{ego}}$ and the two reference frames $\mathbf{P}_{\text{ref}}$. Finally, $\mathbf{P}_{\text{ego}}$ serves as the PE of query while $\mathbf{P}_{\text{ref}}$ serves as PE of key within the cross attention of ControlNet (Fig. 5 (a)(b)):

$$\text{H}'_{\text{cur}} = \text{Attn}(Q = \text{H}_{\text{cur}} + \mathbf{P}_{\text{ego}}, K = \text{H}_{\text{ref}} + \mathbf{P}_{\text{ref}}, V = \text{H}_{\text{ref}}) \tag{4}$$

which enables the current frame to find the correspondence in reference images so that it can follow the static background and generate the transformed pixels.

By utilizing retrieval-based conditions, the street scene on both sides of the ego vehicle is deterministic and ReGen4AD is only responsible for generating coherent images. However, this introduces another train-val gap challenge. **During training, if we simply retrieve the nearest images, they would always be the preceding and following frames within a small distance range while during inference, E2E-AD agents could behave differently from experts and thus the distance to reference images could be far,** shown in Fig. 5 (c). As a result, the training would cause the model to overly rely on references and collapse in the large deviation situations during inference.

To address the issue, we propose to let the model see reference images in a wide range of distance during training. Specifically, for each training sample, we employ a hierarchical sampling technique where reference images are selected out of one of the three distance intervals (2m-5m, 5m-10m, or 10m-15m) based on a pre-assigned probability to balance the sampling. In this work, we assign the probability to be (0.1, 0.3, 0.6), significantly increasing long-distance samples compared with simple nearest frames sampling, as in Fig. 6. The proposed sampling makes the model adapt to highly divergent reference images, effectively narrowing the train-val gap and notably improving the generation quality and model's numerical stability. Fig. 7 gives an example of long-range deviation and generation results under different training strategies.

**Changing Lane to the Left**  **Keep Straight**  **Changing Lane to the Right**

Figure 9: **Closed-Loop Interactive Simulation in nuPlan.** Generated image sequences under three different E2E-AD agent behaviors.

Table 3: Open-Loop Planning Ablation

| Temporal Consistency | Retrieval Ref | FID↓ | Detection NDS↑ | Planning avg.L2(m)↓ |
|---|---|---|---|---|
| ✗ | ✗ | 21.06 | 21.80 | 1.19 |
| ✓ | ✗ | 14.04 | 25.75 | 1.17 |
| ✓ | ✓ | **10.95** | **33.04** | **1.15** |

Table 4: Noise & Gaussian Blur Ablation

| Noise Modulation | Gaussian Blur | FID↓ 0.5s | 1s | 1.5s | 2s |
|---|---|---|---|---|---|
| ✗ | ✗ | **12.68** | 16.09 | 23.77 | 31.79 |
| ✓ | ✗ | 14.87 | 17.64 | 18.72 | 19.58 |
| ✓ | std=1 | 14.40 | **16.84** | **18.01** | **18.67** |
| ✓ | std=2 | 15.55 | 18.14 | 19.11 | 19.79 |

Table 5: Closed-Loop Planning

| Methods | BEVFormer NDS↑ | mAP↑ | R-CLS ↑ |
|---|---|---|---|
| Log-Replay | 0.05 | 0.03 | 27.24 |
| No Ref & No Prev | 23.31 | 12.21 | 28.56 |
| ReGen4AD | **28.23** | **17.23** | **30.49** |

Further, we observe that the generation process might overly rely on reference images and thus leads to bad generalization ability. To alleviate the issue, we apply classifier-free guidance (Ho & Salimans, 2022) (CFG) to reference image, dubbed reference CFG, where we randomly substitute reference images with empty images during training. Similar to standard CFG, at each denoising step during inference, we weighted sum the two predicted noise, one with reference images and one without. As shown in Fig. 8, reference-CFG alleviates model's reliance on reference image, resulting in more authentic foreground objects generation.

## 4 EXPERIMENT

### 4.1 EXPERIMENTAL SETTINGS

We conduct detection, mapping and open-loop planning evaluation on **nuScenes dataset** (Caesar et al., 2020) and closed-loop planning evaluation on **nuPlan dataset** (Karnchanachari et al., 2024). We use the nuScenes's official train-val-test split and nuplan mini split (around 5 times larger than nuScenes) due to limited computational resource.

We evaluate the generation quality with Frechet Inception Distance (FID) and Frechet Video Distance (FVD). The layout conformity is evaluated through object detection and BEV segmentation on generated images with widely used baselines BEVFormer (Li et al., 2022) and BEVFusion (Liu et al., 2024) (camera branch). To evaluate the temporal consistency, we employ state-of-the-art streaming perception models StreamPETR (Wang et al., 2023a), which features the reuse of agent queries from previous frames, so improved perception scores indicate better temporal consistency. To evaluate the influence on planning, we employ UniAD (Hu et al., 2023b) and VAD (Jiang et al., 2023) for open-loop and closed-loop evaluation respectively.

Please refer to supplemental materials for more implementation details and metric introduction.

### 4.2 QUANTITATIVE ANALYSIS

**Generation Quality and Controllability**. We evaluate ReGen4AD with nuScenes validation dataset. As shown in Tab. 1, it outperforms baselines in generation quality, yielding notably lower FID and FVD scores. For controllability, better perception and segmentation scores are achieved on ReGen4AD-generated images, indicating better generation precision for objects and map elements.

**Temporal Consistency.** ReGen4AD can yield consistent sensor image sequences over a long horizon, which is crucial for perception models with high temporal reliance. As shown in Tab. 1, perception scores with StreamPETR and FVD are notably better than the baseline method. This demonstrates the effectiveness of our method in improving temporal consistency.

**Open-Loop Planning.** As shown in Tab. 2, UniAD performs the best when evaluated on ReGen4AD generated sequences than on other baselines under nuScenes open-loop evaluation protocol.

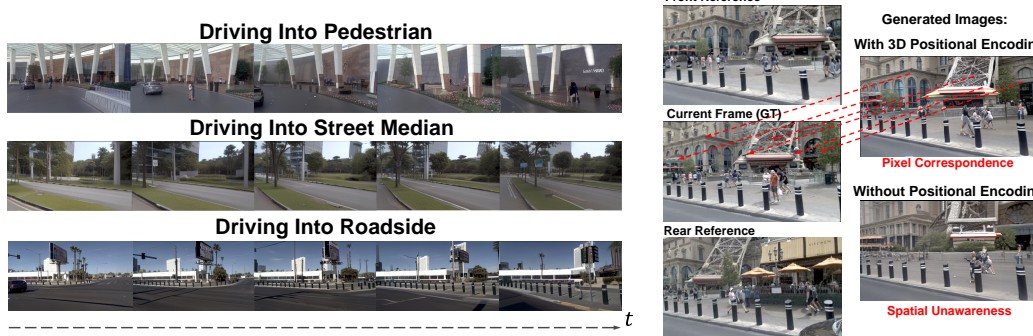

Figure 10: **Generalizability.** Out-of-distribution generation results under scenarios absent in the training dataset.

Figure 11: **Effect for 3D Positional Encoding.**

**Closed-Loop Planning.** We integrate the ReGen4AD framework into nuPlan for **closed-loop reactive simulation**. We adopt the Val14 evaluation split (Dauner et al., 2023). However, since only 10% scenes in nuPlan have sensor data, we filter 10 full clips from each of the 14 scenarios and report R-CLS score and perception score. We use two simple image acquisition methods to serve as baselines, log replaying (collecting images with corresponding timestep from recorded ego trajectories) and static frame generation with no previous prior or reference frames.

As shown in Tab. 5, ReGen4AD's perception scores and R-CLS of VAD are notably higher than baseline methods, indicating sensor images' great adherence to bbox-level simulation environment and the efficacy of our methods. During the experiment, we find that driving scores of VAD tend to drop to zero at the early stage of simulations. This is because VAD, which adopts an imitation learning paradigm, is not capable of coping with long horizon closed-loop simulation, which is aligned with previous findings (Cheng et al., 2023). We provide more case studies for VAD planning ability in supplementary material.

### 4.3 QUALITATIVE ANALYSIS AND ABLATIVE STUDY

**Closed-Loop Interactive Simulation**. As in Fig. 9, ReGen4AD is able to generate high-fidelity images with different behaviors of E2E-AD agents under the same scene.

**Generalizability**. As in Fig. 10, ReGen4AD can generate authentic sensor images even under scenarios absent in training dataset, demonstrating the rich real-world prior knowledge in the pretrained diffusion model.

**Designs of ReGen4AD.** In Tab. 3, we ablate the two proposed simulation oriented designs and results show that task performance improves along with the modules we add, demonstrating their effectiveness.

**Noise Modulation with Gaussian Blurring** As shown in Tab. 4, directly encoding previous images leads to fast deterioration while with noise modulation and a proper level of Gaussian noise added, generalization quality remains stable during autoregressive generation process.

**3D Positional Encoding**. As shown in Fig. 11, with the explicit spatial transformation information provided by 3D PE, generated images exhibit pixel-level correspondence.

## 5 CONCLUSION AND OUTLOOK

We present ReGen4AD, a simulation-oriented generative framework that enables reactive closed-loop evaluation for end-to-end driving models. We integrate ReGen4AD into nuPlan simulator to develop a reactive simulation platform. We prove the efficacy of the simulation-oriented designs through thorough experiments.

**Limitation and Future Work.** Since ReGen4AD mainly focuses on consistent agent and background generation, we doesn't incorporate any diffusion acceleration techniques and the generation of traffic lights and pedestrians remains underdeveloped as with most existing works in this area. Future work will prioritize fast generation and consistency in vulnerable road users.

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

## A APPENDIX

### A.1 LDM AND NUPLAN

**Latent Diffusion Model with Control**. ReGen4AD is based on latent diffusion model (LDM) (Rombach et al., 2022b). LDM consists of two components: a variational autoencoder (VAE), which compresses input images to latent space with an encoder $z = E(I)$ and reconstructs latent features to image space with a decoder $I = D(z)$, and a 2D U-Net, which is trained by predicting the noise added to latent features at timestep $t \in (1, 2, \ldots, T)$. Training loss for LDM is:

$$\mathcal{L}_{\text{LDM}} = \mathbb{E}_{\epsilon_t \in \mathcal{N}(0,1), t \in \mathcal{U}[0,T], c} \left[ ||\epsilon_t - \epsilon_\theta(z_t; t, c)||^2 \right] \tag{5}$$

where $z_t$ is the noisy latent at timestep $t$, $\epsilon_\theta$ is the noise prediction network to be trained, $c$ is the control for conditional generation. During inference, LDM generates images by iteratively removing U-Net-predicted noise from randomly sampled Gaussian noise for $T$ steps. For fair comparisons

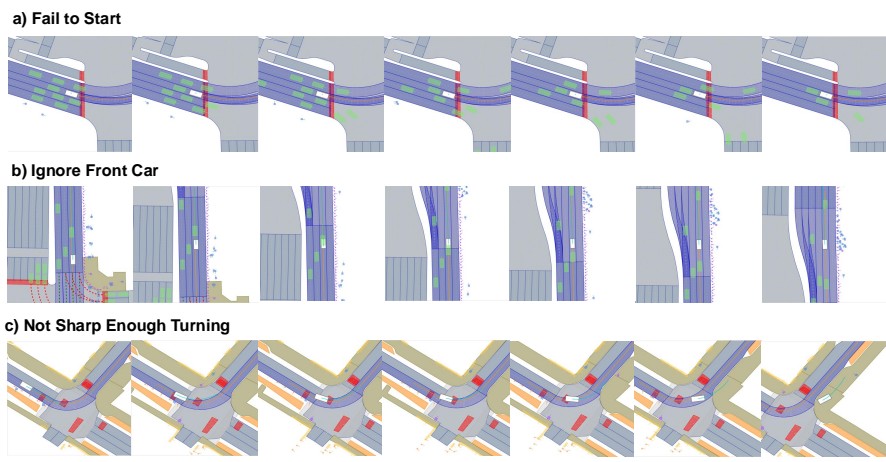

Figure 12: **Typical Failure Cases of VAD** We select three typical failure cases of VAD: failing to start, accelerating when there are cars in the front and failing to take turns. The white box is the ego car; green boxes are other driving cars; the green line is the planned ego trajectory and the orange line is logged expert trajectory.

with existing works (Gao et al., 2024a), we adopt the pretrained Stable Diffusion v1.5 as our base LDM.

Aside from using text prompt guidance in the original LDM, ReGen4AD also incorporates pixel-space guidance using ControlNet (Zhang et al., 2023). ControlNet creates a trainable copy of the U-Net encoder. The outputs from each layer of the ControlNet are added to the outputs of the corresponding layer in the original U-Net encoders. ControlNet and U-Net are connected via zero-conv module to prevent random noise at the early stage of training.

**nuPlan Simulator and Benchmark**. nuPlan (Karnchanachari et al., 2024) is a widely used reactive closed-loop planning benchmark based on large-scale real-world data. nuPlan divides a long real-world driving journey into smaller, manageable driving scenarios. Each scenario has high-level navigation information such as goal points and route plans. It also contains sensor data collected along expert trajectories.

## A.2    IMPLEMENTATION DETAILS

### A.2.1    TRAINING AND INFERENCE

ReGen4AD is initialized with SDv1.5 (Rombach et al., 2022a) official pretrained weights. At the training stage, we optimize our renderer for 50k steps with a total batch-size of 114 on nuScenes, and for 140k steps with a total batch-size of 64 on nuPlan. The learning rate is set to be $1e-4$ and cosine-annealing scheduler is employed with warm-up steps to be 3k steps. We assign the dropout rate of retrieved reference images to be $0.2$, and Reference CFG's guidance scale is set to 2 at inference time.

During inference, following (Gao et al., 2024a), images are sampled using UniPC (Zhao et al., 2023) schedulers for 20 steps. All sensor images are sampled at a spatial resolution of $400 \times 224$ and then upsampled to the original size with bicubic (Keys, 1981) interpolation, namely $1600 \times 900$ for nuScenes and $2000 \times 1200$ for nuPlan. We use the official UniAD pretrained weight on nuScenes and train an 8-views version of VAD on nuPlan dataset. Although the simulation frequency of nuPlan simulator is 10Hz, we follow the convention in the community and set the inference frequency of VAD to be 2Hz, and use the most recently predicted trajectory to propagate world state at the intermediate frames.

### A.2.2 METRICS

For evaluation on nuScenes (Caesar et al., 2020) dataset, we adopt the official detection evaluation protocol. The mean Average Precision (mAP) metric utilizes ground-plane center distance instead of 3D IoU for prediction-ground truth alignment. The framework further incorporates five true positive (TP) metrics: ATE (translation), ASE (scale), AOE (orientation), AVE (velocity), and AAE (attribute errors). These components are weighted in the NuScenes Detection Score (NDS) to holistically quantify detection performance.

For closed-loop evaluation on nuPlan (Karnchanachari et al., 2024) dataset, we employ **CLS** (Closed-Loop Score) defined by the official nuPlan challenge. CLS is a scenario-based metric, which comprehensively combine multiple aspects of driving performance assessments including drivable area compliance, collision time, progress along the driving direction, comfort, etc.

### A.2.3 SIMULATION SETUP

We base the behavioral controller of our simulation framework on nuPlan (Karnchanachari et al., 2024) simulator. At each timestep $t$, ReGen4AD generates surrounding sensor images. The E2E driving agent uses the sensor image to plan future trajectory without any kinematic feasibility requirement. Following nuPlan simulator, our behavioral controller propagates ego motion in a two-stage manner. We first use a Linear Quadratic Regulator(LQR) to find the optimal control policy and then feed it into a kinematic bicycle model to propagate simulation state.

Other driving agents respond to ego's behavior in a reactive way following the Intelligent Driver Model(IDM) policy. IDM agents are parametrized using logged agents' kinematic states (pose, velocity) and also strict lane-centerline adherence based on map geometry while avoiding possible collision with ego agent.

### A.3 ANALYSIS ON VAD PLANNING PERFORMANCE

We provide some typical failure cases of VAD in Fig. 12. As suggested by Fig. 12 (a) and (b), VAD is hard to start from static states, and is unable to slow down even when there are slow cars in the front. This proves the findings (Cheng et al., 2023; Zhai et al., 2023) that imitation based driving models are likely to take shortcuts from current kinetic states during training, developing an overly reliance on ego states while ignoring other information during inference.

In Fig 12 (c), VAD's planning results are very close to ground-truth trajectory at the early stage of a left turn. However, the ego car gradually deviates from the original route because of accumulated errors and the model can't adapt to the changes and hit the roadblock. This result intermediately reflects that the now commonly-used open-loop protocols are not capable of evaluating model's true driving ability.

### A.4 MORE QUALITATIVE RESULTS

### A.4.1 GENERATION QUALITY

As shown in Fig. 13, ReGen4AD is capable of generating high-fidelity panoramic images under diverse driving scenarios.

### A.4.2 CONTROLLABILITY & SPACIAL CONSISTENCY

We demonstrate the controllability of ReGen4AD by removing all object bounding boxes in the same driving scenarios, as shown in Fig. 14. For each scenario, we generate two sets of images, one with bounding boxes and one without. Despite the radical changes of object control signals, ReGen4AD is able to achieve high spatial consistency at the background level and removes all foreground objects in the scenario (including driving cars and pedestrians), demonstrating the efficacy of our designs for the generative renderer.

### A.4.3 MORE INTERACTIVE SIMULATION VISUALIZATION

We provide more interactive simulation results in Fig. 15. For each driving scenario, we let the ego driving agent conduct two different behaviors. ReGen4AD ensures great spatial-temporal consistency, providing a coherent simulation environment.

### A.4.4 VIDEO RESULTS

Please refer to our supplementary video for clearer visual demonstrations of the generated results.

### A.5 ETHICS STATEMENT

The primary application of this work is to improve the safety and reliability of autonomous vehicles. By providing a high-fidelity, controllable simulation environment, we aim to accelerate the development and validation of safe driving behaviors, ultimately contributing to the reduction of traffic accidents.

The model was trained using publicly available driving datasets that have been anonymized to protect personal information and privacy. No new data was collected for this study.

We recognize that generative video models could be used to create deceptive content. ReGen4AD is designed as a specialized tool for AD simulation and is not intended for general-purpose video generation. We encourage responsible use within the research and engineering communities.

### A.6 REPRODUCIBILITY STATEMENT

All information necessary to reproduce our results—including a detailed methodology and experimental setup—is contained within this paper. The accompanying code, pre-trained models, and configurations will be made publicly available upon publication.

### A.7 USE OF LLM

We use Large Language Models(LLM), specifically Gemini, as a writing assistance tool to help polish the manuscript, improving grammar, clarity, and readability of the text. We confirm that the core contribution in the manuscript, including the scientific ideas and experimental results, are entirely our own.

nuScenes

nuPlan

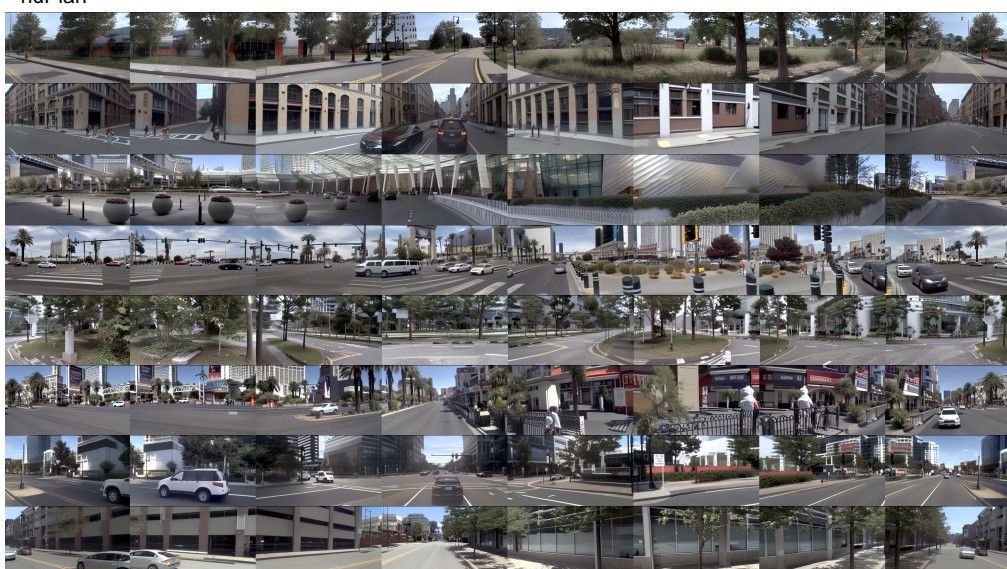

Figure 13: **Generated Images In NuScenes and NuPlan.** ReGen4AD is capable of generating diverse driving scenarios with high fidelity.

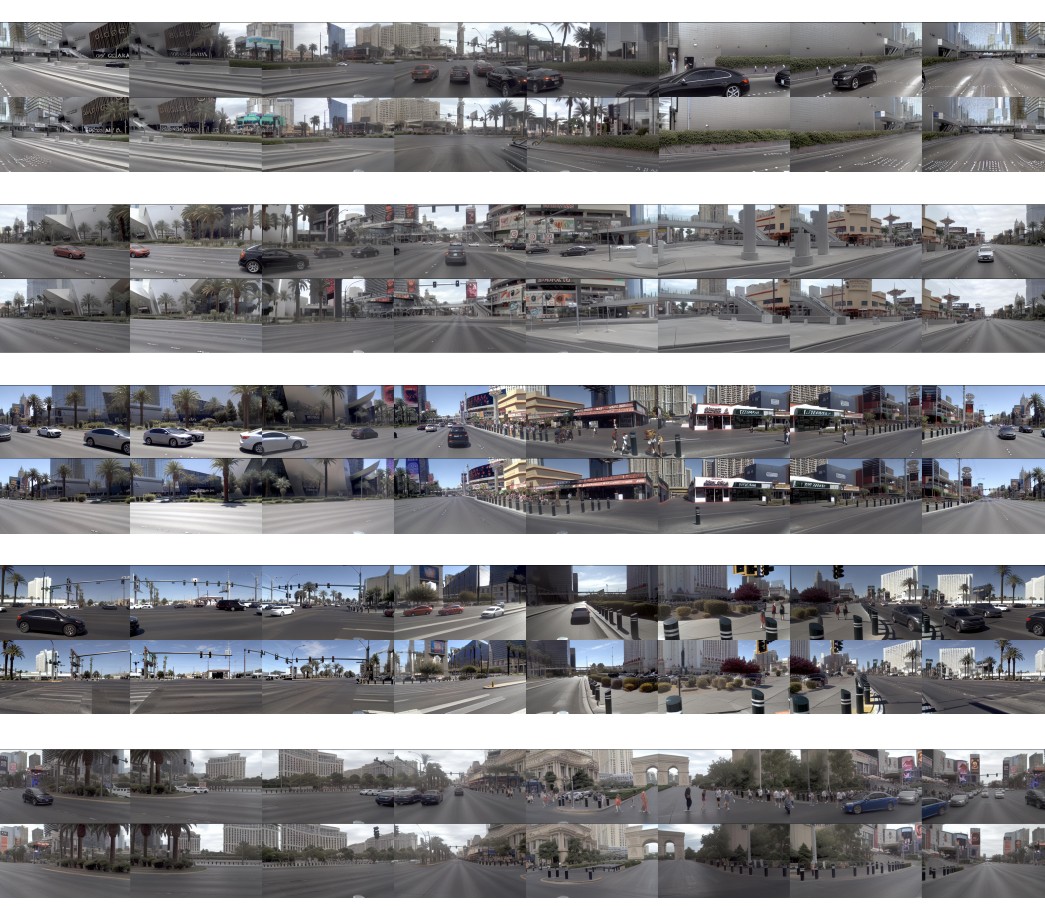

Figure 14: **Controllability and Spatial Consistency** For each set of images, the upper row is generated with object bounding boxes and the lower row without. ReGen4AD abides strictly by the control signals while maintaining high background consistency.

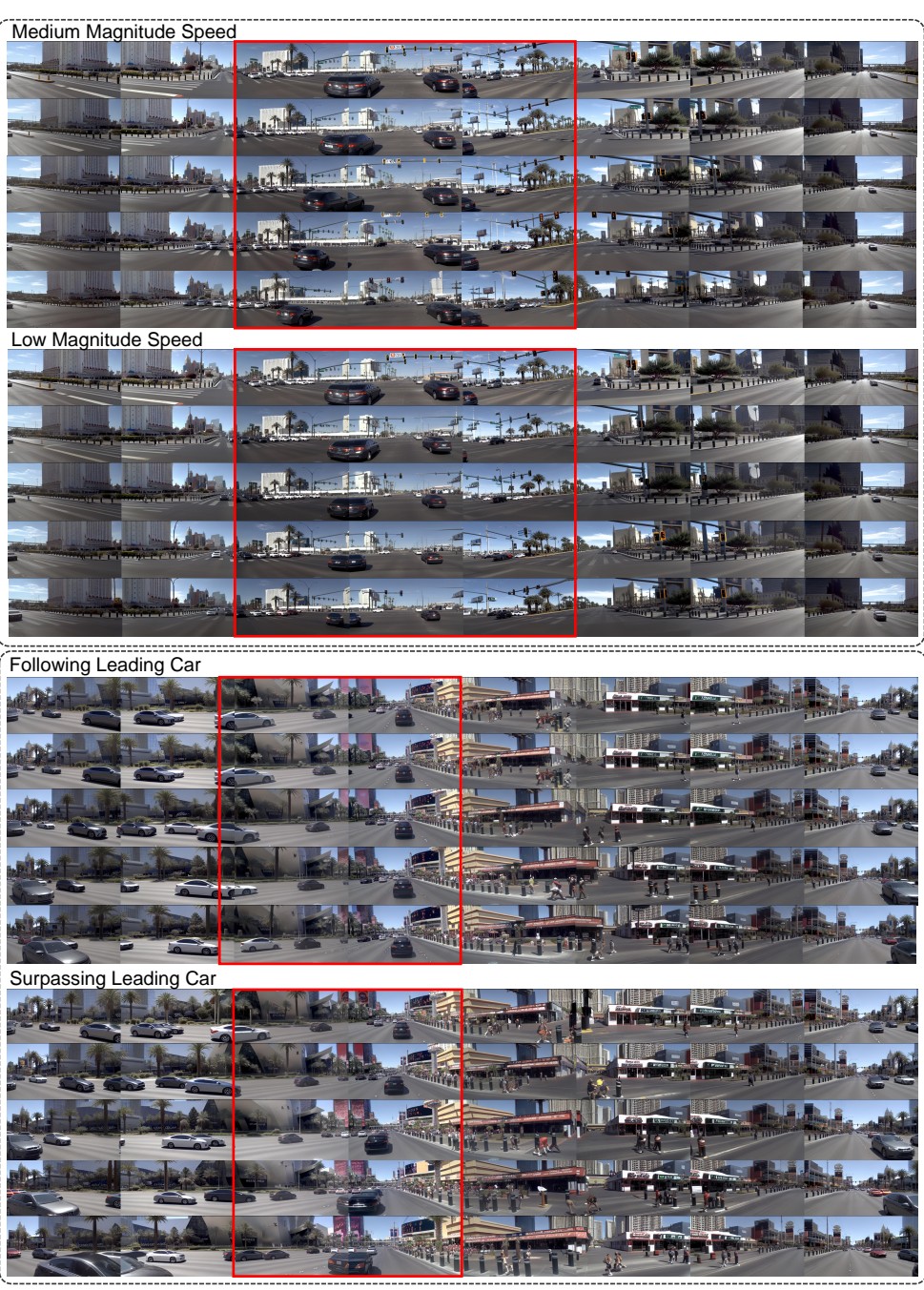

Figure 15: **Interactive Simulation Result**. Views with most conspicuous differences are highlighted with red boxes.

