# OpenReview forum: "ReGen4AD: Retrieval Based Online Video Generation for Reactive Autonomous Driving Simulation"
_ICLR.cc/2026/Conference — ICLR 2026 Conference Withdrawn Submission_

### Official Review · Reviewer_ehZN · 2025-10-30

**Soundness:** 3
**Presentation:** 3
**Contribution:** 2
**Rating:** 2
**Confidence:** 4

**Summary:**

They propose an interactive video generation method specifically designed for interactive simulation or closed loop evaluation. The main framework in the paper is designed around a video generation approach that is more suitable for closed-loop simulation. To achieve the stepwise interactive characteristics of simulation, they used an autoregressive frame-by-frame generation method for model design. To mitigate the cumulative errors in autoregressive generation, a common method of adding Gaussian noise to the conditional frames is used. To enhance spatial consistency, a method is proposed that indexes images with nearby spatial positions in the dataset to be used as reference frames.

**Strengths:**

1. The video generation method designed for closed-loop simulation scenarios aligns well with the input-output framework of closed-loop simulations, such as controllability. Additionally, the use of an autoregressive frame-by-frame generation paradigm and the idea of enhancing the generation effect by retrieving existing frames from a database are overall quite reasonable.
2. The logic of the paper is generally clear, and the illustrations in the method section are also very well-drawn.

**Weaknesses:**

1. There are issues with the writing, such as incorrect citations on line 141, and the mention of "drivedreamer2" and "drive-wm" on line 146, which I believe should be classified under "generation with layout-controlled video." The title of Figure 8 is also incorrect.
2. The main contribution of the paper lies in introducing a retrieval-based method for generating historical data in closed-loop simulation tasks. This works well as a condition for static backgrounds, but from the current framework's perspective, the previous and next frames of dynamic objects serve as harmful conditions for the current frame.
3. There is a lack of comparison with other closed-loop simulations using generative methods, such as Driving Arena [1] and Drivingsphere [2]. Even if it’s inconvenient to compare directly, the differences should be clearly explained.
4. The work of frame-by-frame generation, which can also achieved by layout generation methods like Panacea, using multiple frames as conditions to generate the last frame. Please clarify this comparison in the paper.
5. The baselines mainly compared in the paper are quite outdated. While this does highlight the inferior video generation quality, this is somehow reasonable because your model operates under a frame-by-frame generation setting, which differs from video generation. Perhaps you could try comparing video generation models like Panacea by modifying them to frame-by-frame generation as described in Disadvantages 4.
6. There are also specialized works on frame-by-frame controllable generation, such as Glad [3], which are not compared in the paper.
7. There is a lack of comparison with previous video generative methods (open-source model like DreamForge-DiT in Drivearena) applied to closed-loop simulation in terms of R-CLS. Overall, the experiments should be more focused on closed-loop simulation rather than open-loop validation. Since you aim to propose a better video generator for closed-loop simulation to demonstrate its effectiveness, currently, there is only a simple Table 5.

[1] Yang, Xuemeng, et al. "Drivearena: A closed-loop generative simulation platform for autonomous driving." Proceedings of the IEEE/CVF International Conference on Computer Vision. 2025.
[2] Yan, Tianyi, et al. "Drivingsphere: Building a high-fidelity 4d world for closed-loop simulation." Proceedings of the Computer Vision and Pattern Recognition Conference. 2025.
[3] Xie, Bin, et al. "Glad: A streaming scene generator for autonomous driving." arXiv preprint arXiv:2503.00045 (2025).

**Questions:**

1. Will the code be open-sourced, and will you be able to provide a certain amount of video demonstrations in the rebuttal period?
2. What is the time taken to generate one image?
3. Is the ego steering angle not included in the ego-to-global matrix? Is "coordinates" not included in the ego-to-global matrix either? Where is the ego velocity used?
4. The specific training process of the model needs to be clarified. Was it trained on the nuScenes train split and then trained on nuPlan, or were they trained separately using one model?

---

### Official Review · Reviewer_Bq3m · 2025-10-30

**Soundness:** 3
**Presentation:** 2
**Contribution:** 2
**Rating:** 4
**Confidence:** 3

**Summary:**

This paper proposes ReGen4AD, a retrieval-based online video generation framework aimed at addressing the sensor image generation problem in closed-loop reactive simulation for autonomous driving (AD). Its core design decouples behavioral control from sensor rendering: an independent behavioral controller is used to simulate the reactions of surrounding agents, enabling the generative model to focus on image fidelity, control adherence, and spatiotemporal consistency.To achieve temporal consistency, a noise-modulated temporal encoder with Gaussian blurring is designed to alleviate distribution shifts and error accumulation in autoregressive generation. For spatial consistency, a retrieval mechanism based on an offline street-view database is introduced, selecting the spatially nearest front and rear reference frames. The spatial relationship between the target frame and the reference frames is explicitly modeled using 3D relative position encodings. Meanwhile, hierarchical sampling and classifier-free guidance (reference CFG) are employed to mitigate the potential over-reliance on reference images.

**Strengths:**

The ReGen4AD, for the first time, an online video generation framework designed for closed-loop, reactive autonomous driving (AD) simulation. The proposed framework effectively addresses key shortcomings of existing methods—while frame-level generation suffers from poor temporal consistency, video-level generation lacks stepwise interactivity. By decoupling behavior control from rendering, the framework enables high-frequency interaction with end-to-end (E2E) AD models, thus aligning well with the closed-loop “perception–planning–control” paradigm observed in real-world driving scenarios. The proposed retrieval mechanism, together with 3D positional encoding, leverages real background priors from an offline street-view database to mitigate hallucinated artifacts commonly produced by generative models. Qualitative results demonstrate that this approach preserves pixel-level spatial correspondence. Moreover, the hierarchical sampling strategy effectively reduces the reference-frame distance gap between training and inference.The proposed network achieves state-of-the-art performance across multiple benchmark datasets.

**Weaknesses:**

Although the proposed method demonstrates certain performance advantages, I find its novelty to be relatively limited. Specifically, Section 3.2 introduces only a simple noise-augmentation strategy for training samples, while Section 3.3 follows the paradigm of existing methods and  does not constitute a substantive contribution. In addition, the design and analysis of the ablation studies in Section 4.3 are not sufficiently thorough.

**Questions:**

- Could the authors clarify how the choices of reference frame distance and sampling probability were determined? Specifically, was there an empirical study or theoretical rationale guiding these selections? It would be helpful to understand whether these two parameters have a significant impact on the model’s performance. Furthermore, it would be insightful to compare the proposed strategy with a simpler baseline, such as always selecting the nearest frame. How much performance improvement does the proposed method provide over this simpler approach? Including an ablation study or sensitivity analysis on these parameters could provide stronger evidence for the effectiveness of the chosen settings and help readers understand the robustness of the method.
- Is the Gaussian Blur in Table 4 already the optimal value? And at 1 s, is the FID lowest when the standard deviation is 1?
- It appears that the number of decimal places for the performance metrics in Tables 1 and 2 is inconsistent. For clarity and readability, it is recommended to standardize the decimal places, at least within each column, so that values are easy to compare at a glance. A thorough check of the number formatting throughout all tables in the manuscript would ensure a more professional presentation and avoid potential confusion for readers.

---

### Official Review · Reviewer_zT3f · 2025-11-01

**Soundness:** 4
**Presentation:** 4
**Contribution:** 4
**Rating:** 6
**Confidence:** 4

**Summary:**

The paper introduces ReGen4AD, a novel framework for interactive and controllable video generation tailored for closed-loop evaluation of end-to-end autonomous driving (AD) systems. Unlike existing generative models that produce entire video sequences at once or lack interactivity, ReGen4AD decouples sensor rendering from behavioral simulation, enabling step-by-step interaction with AD models.

**Strengths:**

- ReGen4AD is the first framework designed specifically for interactive, closed-loop evaluation of AD systems, addressing a critical gap in current generative modeling for driving simulation.
- The noise modulation and Gaussian blurring strategy effectively tackles the autoregressive distribution shift problem, a common issue in sequential generation.
- The retrieval mechanism with hierarchical sampling and 3D positional encoding enhances spatial consistency and reduces over-reliance on reference images.
- Extensive experiments on both open-loop (nuScenes) and closed-loop (nuPlan) benchmarks validate the method’s superiority in generation quality, temporal coherence, and planning performance.

**Weaknesses:**

- The paper does not address the inference speed of ReGen4AD, which is critical for real-time closed-loop simulation. Latency comparisons with baseline methods are missing.
- While compared with other generative models, ReGen4AD is not evaluated against traditional non-generative simulators (e.g., CARLA) in terms of realism or simulation-to-real gap.

**Questions:**

- How does ReGen4AD compare with non-generative simulators like CARLA in terms of rendering quality and behavioral realism?
- Have you tested the framework for long-horizon simulations? Does the quality degrade over time despite the proposed noise modulation?

---

### Note · Authors · 2025-11-12

I have read and agree with the venue's withdrawal policy on behalf of myself and my co-authors.